# Diagnostic and Prognostic Value of Clinical Scoring and Lung Ultrasonography to Assess Pulmonary Lesions in Veal Calves

**DOI:** 10.3390/ani13223464

**Published:** 2023-11-09

**Authors:** Julia Hoffelner, Walter Peinhopf-Petz, Thomas Wittek

**Affiliations:** 1PFI Dr. VET—The Veterinary OG, 8403 Lang, Austria; walter.peinhopf@dr-vet.at; 2University Clinics for Ruminants, University of Veterinary Medicine Vienna, 1210 Vienna, Austria; thomas.wittek@vetmeduni.ac.at

**Keywords:** bovine respiratory disease, thoracic ultrasound, pulmonary lesions in veal calves, antimicrobial resistance

## Abstract

**Simple Summary:**

Respiratory disease in calves is often accompanied by an extensive usage of antimicrobial drugs. As part of the strategy to reduce antimicrobial resistance by “prudent use of antibiotics”, veterinarians and farmers need to ensure that individual therapy decisions are based on a “prior risk assessment” of the severity and prognosis of the respiratory disease. To achieve this goal, the integration of the on-farm diagnostic tool of ultrasonography and a simple clinical scoring system to evaluate the degree and severity of lung lesions is recommended. Our study highlights the importance of an early diagnosis of respiratory disease and shows that ultrasonography of the thorax is most useful in the early detection of lung tissue changes and should assist in prioritizing calf treatment and reducing the use of antibiotics.

**Abstract:**

This study on veal calf respiratory disease assessed the association between an on-farm clinical scoring system and lung ultrasonography with the postmortem inspection of the lungs. The comparisons allowed the calculation of predictive values of the diagnostic methods. In total, 600 calves on an Austrian veal calf farm were examined at the beginning and the end of the fattening period. Overall, the area under the curve (AUC) for ultrasonographic scores was 0.90 (rsp = 0.78) with a sensitivity (Se) of 0.86. The specificity (Sp) was 0.78, and the positive predictive value (PPV) was 0.74. The AUC for the physical examination was 0.76 (rsp = 0.55) with a Se of 0.64, an Sp of 0.81, and a PPV of 0.69. For the combination of ultrasonography and physical examination, an AUC curve of 0.85 (rsp = 0.69) was calculated. A Se of 0.65 and a Sp of 0.88 with a PPV of 0.73 was calculated. This study concluded that both physical and ultrasonographic examination scoring are reliable examination methods for the detection of lung diseases in veal calves.

## 1. Introduction

Dairy calf pneumonia (DCP) belongs to the bovine respiratory disease complex (BRD) and affects calves younger than six months of age [1]. Because of the negative impact on animal welfare [2] and economic importance, DCP is considered one of the most important calf diseases in the livestock industry [3]. Early detection and prevention of respiratory disease reduces not only direct treatment costs but also indirect costs, for example, poor growth in veal calves and poor reproduction in dairy heifers [4,5,6,7].

Disease management protocols (e.g., treatment and vaccination protocols) [6] and scoring systems [5,8,9,10,11] are considered cost-effective tools to identify and treat clinically sick calves rather than the blanket treatment of all calves in the affected group, hence reducing antibiotic use and potential antimicrobial resistance [12]. Scoring systems code the physical examination procedure for the personnel working with the calves (farm staff), although the accuracy of the assessment of clinical criteria (rectal temperature (T), nasal discharge (ND), eye discharge (ED), cough (CO), head tilt, or ear drop (HAT) and assessment of breathing (AB)) is variable. The subjective perception of the clinical criteria and the absence of a gold standard method for the ante mortem detection of respiratory disease provide two reasons for the limited diagnostic value of this method and the need for greater refinement [13,14,15]. It is assumed that a higher sensitivity of 0.79 with a confidence interval (CI) of 0.66–0.91 and a specificity of 0.94 with a CI of 0.88–0.98 could be reached by ultrasonographic lung examination [16]. The ultrasonographic examination allows visualization of the extent and degree of severity of abnormal lung tissue caused by the inflammatory processes [8,16,17,18]. The ultrasonographic image of healthy lung tissue is represented by artifact lines. These so-called reverberation artifacts occur parallel to the lung surface because of the total reflection of the air. Pulmonary or pleural processes like consolidation, abscesses, or pleural effusion [19] can be visualized by ultrasonographic examination. The standard protocol for the on-farm use of ultrasonographic examination procedure is a fast procedure that takes approximately 60 s per calf for trained observers [20]. Thus, the advantages of ultrasonographic examination procedures are not only seen in a more accurate diagnosis but also practicability. However, the exclusive use of this ultrasonographic examination procedure has not been proven as an appropriate gold standard method for predicting respiratory disease. Indeed, as Buczinski et al. (2014) point out, “the imperfect accuracy of calf respiratory scoring charts and ultrasonography should be taken into account when using those tools to assess BRD status” [18]. In particular, the inability to differentiate between chronic or acute lung disease processes may result in an inaccurate assessment of the BRD status of the calves and the farm. The absence of lung consolidation in the early stage of BRD might be misinterpreted as healthy [21], and in contrast, lesions that are unrelated to BRD, for instance, neoplasms, might be noticed as BRD disease [21]. In short, the reference point of postmortem lung changes is the gold standard, but even this varies with time. Thus, an examination at entry is not the same as a postmortem picture at slaughter.

The main objectives of this study were first to evaluate the extent and severity of lung diseases clinically and by ultrasound in calves in an Austrian veal farm at two time periods (1. on arrival at the beginning of the fattening period in calves of approx. 6–8 weeks of age and 2. at the end of the fattening period approx. 90–120 days later). Second, these clinical and ultrasonographic data were compared to a visual assessment of lung lesions after slaughter, allowing the evaluation of the diagnostic and prognostic value of these examinations.

Two hypotheses were formulated: First, clinical signs and ultrasonographic findings of lung diseases in veal calves at the beginning of the fattening period are associated with findings of thoracic ultrasonography at the end of the fattening period (approx. 90 days later). Second, the clinical scores and ultrasonographic findings at the end of fattening are well correlated with lung pathology at slaughter.

## 2. Materials and Methods

### 2.1. Animals and Housing

This study was conducted at an Austrian veal farm (Gassner GmbH, 8010 Graz, Rohrbachhöhe 23, Austria). The farm purchases 50 to 70 calves weekly at various cattle markets in the Austrian federal states of Styria and Salzburg. Depending on breed and age, the body weight was variable (medium weight: 90 kg; minimum: 60 kg; maximum: 130 kg). For quarantine purposes, all calves stay for five days in individual boxes in a closed separate section of the farm before they are released and kept in groups of five to six calves per pen. Calves were fed three times per day by an automatic system with milk replacer and were offered muesli (farm-made with varying percentages of barley, oats, and soybeans). The facilities comply with the national requirements for animal husbandry and welfare. After a fattening duration of 76 d on average (Minimum: 41 d; maximum: 130 d), all calves were slaughtered for veal production (Abattoir Weiz, 8160 Weiz, Werksweg 102, Austria). The study was discussed with the institutional ethics and welfare committee and approved by the Austrian Federal Ministry for Education, Science and Research (GZ 2020-0.773.262) in accordance with Good Scientific Practice guidelines and national legislation.

### 2.2. Study Design

Part one of the study started in September 2020. The inter and intra-examination reliability for ultrasonographic examination was tested by two trained observers. To calculate intra-observer reliability, 22 calves were examined by observer one, Julia Hoffelner (JH), and independently evaluated by observer two, Walter Peinhopf-Petz (WP-P), on the same day. To test the intra-observer reliability, results from the two examinations were compared separately for calves and separately examined lung areas (Figure 1).

In part two of the study, 600 calves were examined for BRD both by a physical and ultrasound examination two days after arrival at the farm on entry to the unit by one author (JH) from March 2021 to March 2022. The second identical examination was performed on the same 600 calves the day before slaughter at the slaughterhouse, i.e., the calves were examined upon arrival at the slaughterhouse, and each calf was restrained in a crush to perform the examination procedure.

The selection of the calves for slaughter was made by the farm. Depending on specific slaughter criteria (e.g., body weight), ten to thirty calves were weekly transported to the abattoir. The observer did not know the identity of the calves so the examinations could be taken blindly. After slaughter, JH inspected the lungs and assigned pathological findings to the calves. The study was divided into four seasonal examination cycles (spring: March–May, summer: June–August, fall: September–November, winter: December–February), including 150 calves per season.

All calves were unvaccinated but received antimicrobial therapy (amoxicillin) on the day of entry. Calves that developed respiratory symptoms during the fattening period were treated under veterinary supervision. Treatment decisions were made independently from this examination procedure. Austria has achieved a “free of BVDV” status.

### 2.3. Physical Examination

A physical examination of the respiratory system was performed by applying the bovine respiratory disease scoring system for pre-weaned dairy calves [9,23]. As presented in Table 1, this protocol is based on six key signs: ED (Serous, mucous, or purulent eye secretion; uni- or bilateral), ND (Serous, mucous or purulent nose secretion; uni- or bilateral), HAT (slight uni- or bilateral ear droop or head tilt), CO (impulsive and audible exhalation sound) AB (increased breathing frequency, signs of difficulties in breathing) and T (greater or equal to or less than 102.5 °F).

### 2.4. Auscultation

The AU (auscultation) was additionally performed in standing calves using a stethoscope (3 M Littmann Classic III, 3M Medica, St. Paul, MN, USA). The lung field was systemically auscultated from the dorso-caudal aspect to the ventro-cranial aspect of the lung field. The anatomical segmentation of the lung was applied for the ultrasonographic examination and was also used for recording the results of AU. The examiner had to differentiate between normal (Score 0) and abnormal lung sounds (Score 3). Crackles, wheezes, stridores, or an absence of breathing sounds are related to pulmonary diseases and were recorded as abnormal findings [24,25].

Scores 0 or 3 were added to the scoring system for physical examination.

### 2.5. Ultrasonographic Examination

The ultrasonographic examination was performed immediately after the physical examination using a portable linear rectal ultrasound unit (Tringa Linear Vet, Esaote, Genova, Italy). The thorax was not shaved but moisturized using isopropyl alcohol (70%) on both sides to facilitate ultrasonographic visualization, as described by Ollivett and Buczinski (2016) [8]. Based on the examination method of Ollivett and Buczinski (2016), the linear probe 7.5 MHz was initially positioned in the 10th inter-costal space (ICS) and moved from the dorsal to the ventral parallel to the ribs [8]. Placing the probe to the next cranial ICS, every ICS was scanned systemically, beginning on the caudal (diaphragm 6th to 10th ICS) to the cranial (heart) aspect of the lung field starting on the right side of the thorax.

Pathological findings were documented in four areas (L1, L2, R1, and R3) following the anatomical segmentation as presented in Figure 1. The access for imaging the cranial right lung lobe was between the 4th and 1st ICS on the right side (=R1) [8]. The right caudal lung lobe was found between the diaphragm and the 4th ICS (=R3). On the left side, the cranial lobe was scanned between the 2nd and the 4th ICS (=L1). The caudal left lung lobe (=L2) was located directly behind and bounded by the diaphragm (6–10 ICS) [8,22]. A scoring system for different diagnoses was generated to simplify documentation and analysis (Table 2). A distinction between comet tail artifacts (COMT) and consolidation (CON), alveolograms (ALV), atelectases (ATA), and pleural effusions (FLU) was made (Table 2) [8,18] and presented in Figure 2, Figure 3, Figure 4, Figure 5 and Figure 6. COMTs are seen as vertical artifacts, starting from the pleural surface and moving with lung sliding [26]. Due to the frequent occurrence, even in normal, healthy lung tissue, interpretation, and scoring values were supplemented by the presence (Score 2 and 3) or absence (Score 0 and 1) of lung consolidation. Calves without consolidated lung tissue were considered to be healthy. Any other abnormality described in Table 2 is associated with bronchopneumonia [20,27].

### 2.6. Postmortem Inspection of the Lungs

The postmortem examination of the lung was performed after slaughter in the abattoir. The observation of the lungs was included in the meat inspection procedure. Not more than one minute per calve was provided for the inspection of the whole carcass. Before the start of the study, the observer (JH) was trained by official veterinarians at the abattoir to determine the correct classification of lung lesions. The actual examinations begin after several test runs and exercises. Doubtful findings were discussed with experienced veterinarians (specialized and certificated for meat inspection). The lungs were inspected and were scored using a modified scoring system of Leruste et al. (2012) [28]. The scoring system is presented in Table 3. For data analyses, the allocation of lung lesions to the lung lobes was required. Therefore, the same segmentation of the lung as for ultrasonographic examination was used (Figure 1).

### 2.7. Statistical Analyses

For statistical analyses, all data were recorded in Microsoft Excel (version 2310, build 16.0.16924.20054, Microsoft Corporation) spreadsheets. Significant differences between all parameters of physical (CO, ND, ED, AB, HAT, and increased or decreased T and AU) and ultrasonographic examination (COMT/CON, ALV, ATA, and FLU) were calculated and compared between the two examinations and to data of postmortem inspection of the lungs. The relationship between variables was tested with the Chi-square independence test. If the critical value at a significance level of α 0.05 was lower than 3.84, the null hypothesis could be verified.

The associations between examination methods were calculated with Spearman rank correlation coefficients (rsp). The results of all lung areas were pooled to create a total score per calf for each examination method. Scores for ultrasonographic examination were calculated from the total sum of L1 + L2 + R1 + R3 (maximum: 24 points; minimum: 0 points). Scores for the postmortem inspection were generated by calculation of the sum L1 + L2 + R1 + R2 + R3 (maximum: 15 points; minimum: 0 points). For better comparability, total values were converted into percentages. Additionally, associations between findings of ultrasonographic and postmortem examination per lung area (L1, L2, R1, R3) were calculated.

Kappa correlation coefficients were calculated for intra- and inter-observer reliability. The kappa agreement was judged as poor when 0 ≤ ϰ ≤ 0.20, fair when 0.21 ≤ ϰ ≤ 0.40, moderate when 0.41 ≤ ϰ ≤ 0.60, good when 0.61 ≤ ϰ ≤ 0.80, and very good when 0.81 ≤ ϰ ≤ 1 [29]. Statistical significance for all tests was defined as *p* < 0.05.

## 3. Results

### 3.1. Inter and Intra-Observer Reliability

Two experienced observers (the study involved veterinarians JH and WP-P) performed an ultrasonographic examination of both sides of the lungs of 22 calves (88 lung areas). COMT/CON, ATA, ALV, and FLU were detected. Results showed very good agreements for all lung areas (minimum: between R3 ϰ = 0.81; maximum between L1 ϰ = 1, R1 ϰ = 0.83, L2 ϰ = 0.89, *p* < 0.001 for all). Results for intra-observer reliability also show very good agreements for all lung areas (ϰ = 0.90 for L2 and R1, ϰ = 0.95 for R3, and ϰ = 0.91 for L1).

### 3.2. Results of Physical, Ultrasonographic and Post Mortal Examinations

Parameters of all examination findings of physical, ultrasonographic, and postmortem examination are presented in Table 4 and Table 5. Differences between clinical signs at the start and the end of fattening were compared and calculated. Since the study covered a complete year, the seasonal effects were considered. Generally, all recorded clinical signs of respiratory disease tended to decrease during the fattening period—except AB and a “deviation of normal T”. However, the only significant overall difference between examinations at the start and end of fattening was for a decrease in ED (*p* < 0.05). However, significant seasonal differences were found between some of the other parameter examinations (Table 4).

No significant changes in ultrasound examinations over the year were verifiable for ALV, ATA, and FLU. However, COMT/CON were the most frequent findings with statistical differences in seasonal course (Table 5). The highest proportion of calves with ultrasonographic findings of COMTs and CONs was recorded in summer (80.5%). From the perspective of progression analysis (differences between first and second examination), calves arriving in summer and winter do show higher and increasing percentages than calves arriving in transition periods (spring and autumn).

At slaughter, 26.83% of the lungs showed signs of respiratory lung disease. In total, Score 1 was the most frequent. Significant differences were found between the seasons (Table 5).

### 3.3. Development of Ultrasonographic Examination Score from the First to the Second Examination

In Figure 7, the development of the scoring system from the start to the end of fattening is presented. The strongest improvement over time was calculated for Score 2 (*p* < 0.05). An improvement from Score 2 to Score 1 (10.3%) or Score 0 (9.0%) was detected in 19.3% of all calves. In contrast, calves with Score 0 at the start of fattening showed the highest value of deterioration (13.5%). Interestingly, approximately more than half of the calves, in total 6.6% out of 8.8% with Score 3, do show an improvement between the start and end of fattening. An overall change in this score with a similar proportion moving from 3 to 2 (3.2%) and 2 to 3 (3.2%).

### 3.4. Ultrasonographic and Postmortem Findings in Five Lung Areas

The location of (pre-slaughter) ultrasonographic and postmortem inspection findings is summarized in Table 6 and Table 7. In total, findings for both ultrasonographic and postmortem inspection are most frequent in lung area R1. Since lung lobe R2 is not visible for ultrasonographic examination, it is not included in the assessment.

### 3.5. Associations between Examination Methods

A total scoring value per calf was calculated for physical, ultrasonographic, and postmortem inspection. Spearman correlation calculations were calculated for pre-slaughter physical and ultrasonographic examination scores, the combined scoring of both examination methods, and postmortem inspection. Significant positive correlations were noted between physical and ultrasonographic examination rsp 0.70 (*p* < 0.001). For ultrasonographic and postmortem inspection, correlations of rsp 0.78 (*p* < 0.001) were calculated. The correlation between physical examination scores (including auscultation) and postmortem scores was also positive (rsp = 0.54; *p* < 0.001). An rsp of 0.71 (*p* < 0.001) for a combined scoring of physical and ultrasonographic examination methods was calculated. Ultrasonographic and postmortem scoring values in all examined lung areas were also positively correlated (*p* < 0.001). A moderate rsp was calculated for L1 (rsp = 0.56), L2 (rsp = 0.42) and R3 (rsp = 0.41). A high correlation of rsp = 0.81 was generated for R1.

Overall, the area under the receiver operating characteristics (ROC) for ultrasonographic scores as a predictor for lung lesions at postmortem inspection was 0.90 with a sensitivity of 0.86 (95% CI, 0.68–1.04), a specificity of 0.78 (95% CI, 0.65–0.91) and a positive predictive value (PPV) of 0.74. The cut-off value was calculated at 16.7%, meaning calves reaching ≥4 points out of 24 points were considered to be ill. The receiver operating curve for the physical examination was 0.76. The optimal cut-off value was 30.0% with a sensitivity of 0.64 (95% CI, 0.50–0.78), a specificity of 0.81 (0.66–0.96), and a PPV of 0.69. Calves with ≥6 out of 20 points were considered to be ill. For the combination of ultrasonography and physical examination, a receiver operating curve of 0.85 was calculated. The cut-off value was 12.0%. A sensitivity of 0.65 (95% CI, 0.53–0.77) and a specificity of 0.88 (95% CI, 0.79–0.97) with a PPV of 0.73 was calculated. The cut-off value was 27.3%, meaning calves reaching ≥12 points out of 44 were considered to be ill. All ROC curves are presented in Figure 8. This would suggest that around 70.3% of the calves should be treated regarding the ultrasound; the physical examination suggested 45.6%, and the combined scoring of ultrasound and physical examination suggested treatment for 65.6%. In fact, all calves received antimicrobial treatment at the time of arrival, and 19.5% received treatment a second time during fattening. Nevertheless, treatment decisions were based on clinical diagnoses and were made independently from this study procedure.

## 4. Discussion

The ultrasonographic examination of the thorax has been considered a simple and efficient method for diagnosing respiratory diseases [27,30]. The present study confirms this opinion, as we were able to do ultrasound examinations of the thorax of fattening calves on a commercial farm and compare this with a simple slaughter visual scoring system under field conditions.

### 4.1. Assessment of Physical Signs of Respiratory Disease

For this study, physical examination parameters were collected using the bovine respiratory scoring system for pre-weaned calves (including parameters of ND, ED, HAT, CO, decreasing respiration rate, or higher T) [9,23]. This scoring system proved a reliable method for the detection of respiratory disease and its severity, enabling a recommendation on therapy, though just how to choose this depends on the extent of inclusion for antibiotic treatment [10]. This analysis of the scoring parameters on the farm provided comparable statistical information and allowed some determinations of respiratory disease progression. Over the period of one year, a decreasing prevalence of physical examination parameters during the fattening period was found for symptoms of ED, CO, and calves showing a decrease in abnormal breathing, and this is mirrored in the ultrasonographic parameters (COMT/CON) as shown in Figure 2.

Increased frequency of clinical signs of calf respiratory disease in the early stage of fattening has been shown by previous studies of veal calves [28]. Various factors, including transport, mixing increasing number of calves arriving from different farms with different ages, and nutritional and immunity status, have all been considered to be explanatory factors for poor health performance upon arrival on the farm [31,32]. Interestingly, the measurement of T in the present study shows that calves at the time point of the second examination tend to have more often increased body T than calves of the first examination. We believe that this result might be due to the timepoint of the second examination when calves were examined just at arrival at the slaughterhouse, and a crush restraint of each calf was necessary to perform the examination procedure on these larger calves. A temporary increase of T, as reported by Hill et al. (2016), might be the consequence [33].

### 4.2. Assessment of Ultrasonographic Findings of Respiratory Disease

The ultrasound scoring system that we used is a modification of the 6-point scoring system (0–5) described by Ollivett and Buczinski (2016), which depends on ultrasonographic visualization of the degree and extension of lung consolidation and presence of COMTs of the whole lung [20]. Previous studies concluded that the clinical picture of lung emphysema or micro-abscesses on the lung surface is associated with a high occurrence of COMTs [34,35].

In this study, the presence of singular COMTs was considered to be a normal finding if no consolidation was seen (Score 0 and 1). A score of 2 or 3 was given when a combination of COMTs and lung consolidations was seen. In contrast to Score 2, which might indicate an acute viral infection of the lung, Ollivett et al. (2015) and Ollivett and Buczinski (2016) suggested that Score 3 (consolidation of one whole lung lobe) is associated with bacterial bronchopneumonia [20,28]. As we did not do any microbiological examination on the present field study, we are not able to comment on this.

In addition, the inclusion of further ultrasonographic signs (ALV, ATA, or FLU), we believe, should improve diagnostic value. If alveoli are filled with fluid or cellular material, small abscesses can be pictured by hypo- or anechogenic and round structures. If lung parenchyma is highly altered (e.g., widespread consolidations), ATA can be pictured in the form of echogenic lines. Additionally, pleura irregularities or pleural fluid are associated with bronchopneumonia. Because of the rare occurrence of ALV (4.12%), ATA (3.54%), and FLU (0.50%) upon arrival at the farm, no significant conclusions could be drawn for the development of those findings separately. It cannot be excluded that the short time for examination procedures per calf and suboptimal condition at the farm might have resulted in missing minor tissue changes. A very recent infection formed during transport might also be a reason for the low prevalence of tissue alterations, as their occurrence would appear later during the progression of the disease. However, considering the comparison to the second ultrasonographic examination and the postmortem findings, it is reasonable to believe that no serious underdiagnosis has occurred, and the movement between the various COMT grades was a real reflection of the changes in the disease in the calves.

This study indicates that in a well-suited, stable environment and with hygiene standards, improvement rates of ultrasonographic findings between the beginning and the end of fattening can be high. A second reason for the low prevalence of ultrasonographic findings at the end of fattening might be reasoned in early and consequent therapy decisions based on individual therapy plans in the case of illness. Based on the diagnosis of COMTs and lung consolidations, the development of ultrasonographic scores was analyzed. The higher the score at the beginning of fattening, the more likely treatment, and so the higher the improvement compared to the scoring value at the end of fattening. Therefore, for COMT 3, improvement rates of 39.3% could be detected within a year, including all lung areas. Seasonal differences become obvious regarding the prevalence of all examination parameters. The season may influence the micro-climate conditions on the farm [7,36,37] and, therefore, be closely related to the direct influence factors for the occurrence of signs of respiratory disease [38]. They reported a higher risk of development of respiratory diseases in cold seasons (autumn, winter) than in warmer seasons (spring, summer). The analysis of our study shows high percentages of deterioration rates in these transition periods (autumn, winter) for the presence of COMT/CON and physical parameters (e.g., increasing T and an increased AB). Furthermore, as with Brscic et al., (2012), improvement rates of ultrasonographic findings were highest in summer [38]. One explanation might be a lower new infection rate at that time. The discussion of environmental conditions, especially diurnal T, is highly relevant, even in standardized housing conditions [39].

### 4.3. Association between Antemortem Physical and Ultrasonographic Examination Scorings and Scorings of Postmortem Inspection

In the present study, the prevalence of lung lesions at slaughter was lower than the prevalence of abnormal examination parameters before slaughter. This is due to the fact that minimal lung abnormalities cannot be detected by visual inspection, and not all physical signs (CO, ND, ED, AB, AU, HAT, T) are necessarily associated with lung lesions. Nevertheless, severe lung abnormalities were congruent with clinical and ultrasonographic findings. In total, 26.83% of all examined calves had some pathological lung lesions, of which 11.28% showed only mild abnormalities. A higher prevalence of lung lesions was found by Leruste et al. (2012), where 50% of the lungs showed abnormalities at slaughter [28]. Van der Mei et al. (1987) reported that 17% of 2138 veal calves with severe lung lesions at slaughter [40]. In both studies, scoring from 0–3 was noted for the total lung without differentiation between lung lobes. Taking into consideration that the exact localization of lung lesions may give information about the pathogen agents of the disease, this study evaluated the findings of lung lesions in five areas. A similar segmentation was found by Buczinsky et al. (2014) [18]. In the present study, most lesions appear cranioventrally and bilaterally (R1, R2, L1). That may be an indicator of *M. haemolytica* or *H. somni*-induced fibrinous pneumonia [41]. Of course, underlying infections with viruses such as PI3 and RSV also cannot be ignored [42]. As already mentioned, microbiologic examinations were not performed in the present study.

To calculate sensitivity and specificity for the ultrasonographic examination procedure, the postmortem inspection was used as a comparison examination method [28,30]. A lack of accuracy of antemortem examination methods for respiratory disease (especially physical examination parameters) leads to misinterpretations of disease morbidity and uncertainties in making therapy decisions [15]. Our study compared two antemortem examination methods (ultrasonographic and physical examination) to findings of postmortem inspection of the lungs. In our hands, ultrasonographic examination proved a reliable method for the detection of respiratory disease, and this study showed a more significant association with the postmortem findings (rsp = 0.76, *p* < 0.001) than the physical examination (rsp = 0.51, *p* < 0001). No improvement or correlation can be detected using a combined scoring of both examination parameters.

Studies in the past reported that subjective perception of single clinical parameters for untrained observers leads to low inter-observer specificity and sensitivity for this examination method. The consequences might be incorrect therapy recommendations, as cut-off values for a treat or not treat decision depend on the experience of the observer [8]. The additional usage of ultrasonographic examination procedures allows an increasing diagnostic value and objectivity. The objectivity of the present scoring procedure for the ultrasonographic examination procedure was proved by nearly perfect inter-observer reliabilities.

### 4.4. Limitations of the Study

A limitation factor of this study design was that ultrasonographic findings could not be blinded from the general condition and clinical parameters of the calves in this study. However, even if the ultrasonographic examination had been performed by a person who has not done the physical examination, the examiner is working with the calves and might also be biased to a certain extent. Furthermore, the random selection of calves from only one fattening farm provides a low variety of ages. A larger spectrum of different ages might have led to an enlarged variety of diagnoses. On the one hand, most calves in this study showed relatively mild symptoms of respiratory disease, which leads to a limitation of significance for severe cases. On the other hand, the homogeneous distribution of diagnoses prevents an overestimation of test performance created by the comparison of many outliers. Another limiting factor for this study was the absence of histopathological or molecular biological tests as standardized reference methods. The results of both examination methods were compared with the results of postmortem inspection exclusively, which might be an imperfect diagnosis test for the detection of the specific cause of the disease. The strength of this study was the longitudinal study design, including tracing the development of diagnosis between the beginning and the end of fattening. The exact evaluation of improvements or deteriorations of ultrasonographic findings gave the opportunity to analyze the healing rate and draw conclusions about the chronic progression of the disease. Additionally, analysis of data in the seasonal course had a great impact on seasonal variabilities.

Finally, despite the well-established examination procedure and scoring systems for ultrasonographic examinations, the diagnostic value of this examination method has only been evaluated in a limited number of studies. Rademacher et al. (2013) suggested a higher mortality rate of feedlot steers, which were pre-diagnosed with lung consolidations or pleural irregularities at the early stage of fattening (days 0, 3, 6, 9, and 15 of fattening.) [43]. Additionally, a long-term study by Adams and Buczinski (2016) reported higher mortality rates for first-time calving heifers if ultrasonographic findings of high severity (defined by a scoring system 0–4) were found at the age of three months [17]. Nevertheless, the lack of widespread implementation of ultrasonographic examination procedures in calf health programs is because of several factors: costs of the ultrasound device, training in its use, and the additional for examination, but these, as can be seen here, can be relatively easily overcome. However, in addition, so far, it has failed to aid in the differentiation between just recently (acute) or previously (chronic) developed lung lesions. This ability would greatly aid in how one could best target antibiotic use and so help to reduce the usage of antimicrobial therapy. To our knowledge, this is the first report to analyze the development of ultrasonographic and physical examination parameters between the beginning and the end of the fattening period and compares the findings to postmortem lung lesions after slaughter. It shows that the additional input of the ultrasonic findings in assessment is useful, but it now needs to be tested fully (including economically) under field conditions involving treatments and response.

## 5. Conclusions

Both physical and ultrasonographic examination scoring are useful instruments for detecting lung lesions in veal calves. Nevertheless, the limitations of the ultrasonographic examination procedure must be considered when evaluating the disease prevalence of BRD in veal calves. A classification of chronically, acutely respiratory diseased calves and presumed healthy calves might be preferable, especially regarding the prudent use of antibiotics. More research should be done to improve the diagnostic accuracy of ultrasonographic examinations, including records of treatment protocols and disease management protocols.

## Figures and Tables

**Figure 1 animals-13-03464-f001:**
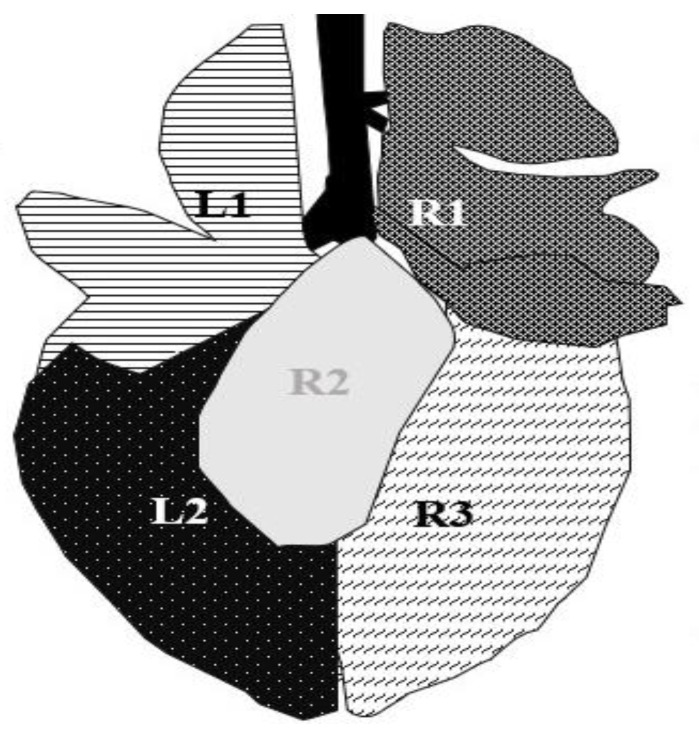
Segmentation of the ultrasonographic examination area, modified from Budras (2002) [22]. The left side of the lung was divided into two (L1 = lobus cranialis sinister, L2 = lobus caudalis sinister) and the right side into three areas (R1 = lobus cranialis dexter + lobus medialis dexter, R2 = lobus accessories, R3 = lobus caudalis dexter). The lobus accessorius (R2) is not accessible for ultrasonographic examination.

**Figure 2 animals-13-03464-f002:**
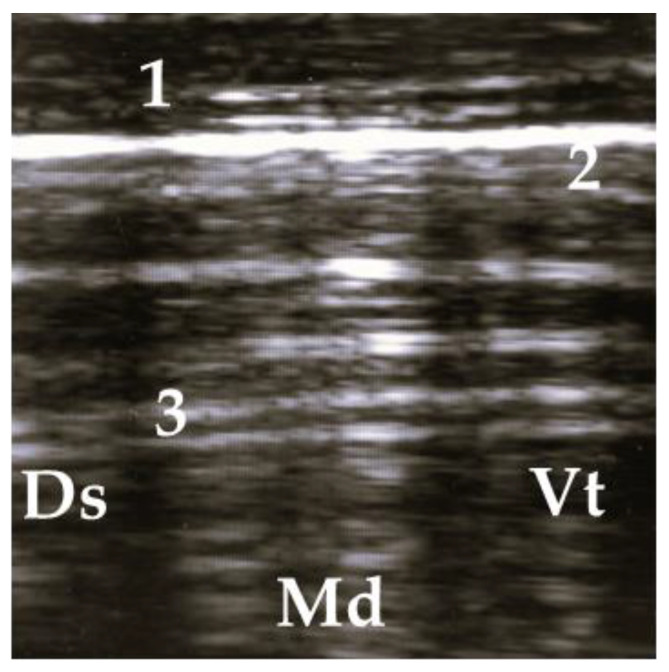
Ultrasonogram of a calf with a normal lung. 1 = thoracic wall, 2 = pleura, 3 = reverberation artifacts; Ds = dorsal, Vt = ventral, Md = medial.

**Figure 3 animals-13-03464-f003:**
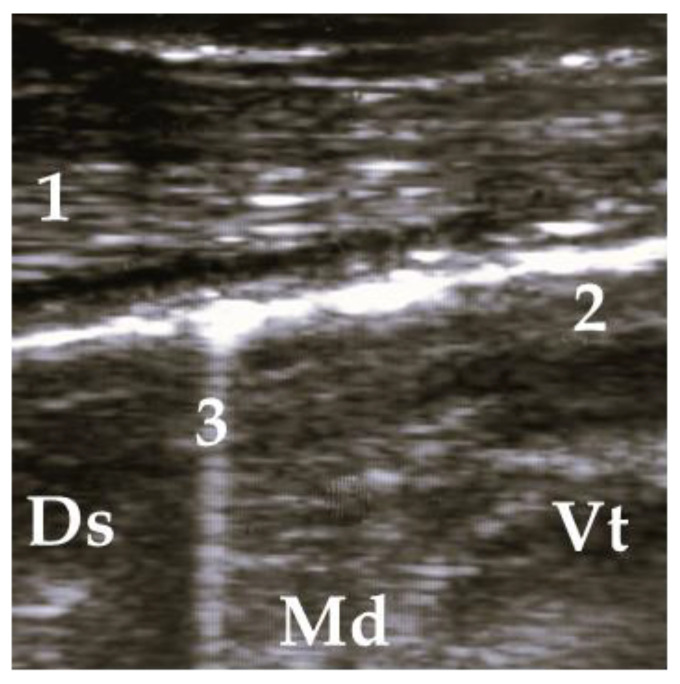
Ultrasonogram of a calf with diffuse allocated COMT, without consolidation. 1 = thoracic wall, 2 = pleura, 3 = comet tail artifact; Ds = dorsal, Vt = ventral, Md = medial.

**Figure 4 animals-13-03464-f004:**
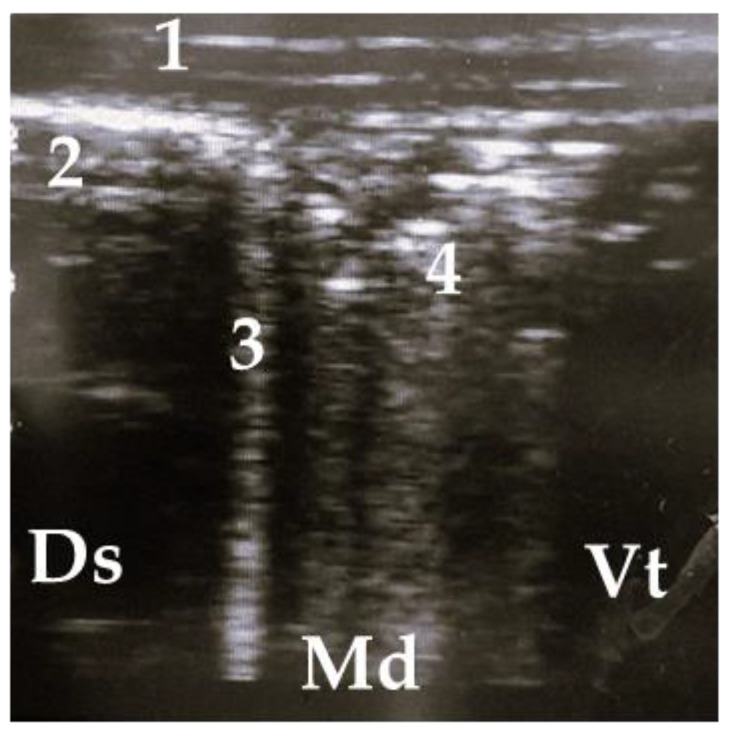
Ultrasonogram of a calf with a small lobar lesion; 1 = thoracic wall, 2 = pleura, 3 = comet tail artifact, 4 = air bronchogram (small consolidations); Ds = dorsal, Vt = ventral, Md = medial.

**Figure 5 animals-13-03464-f005:**
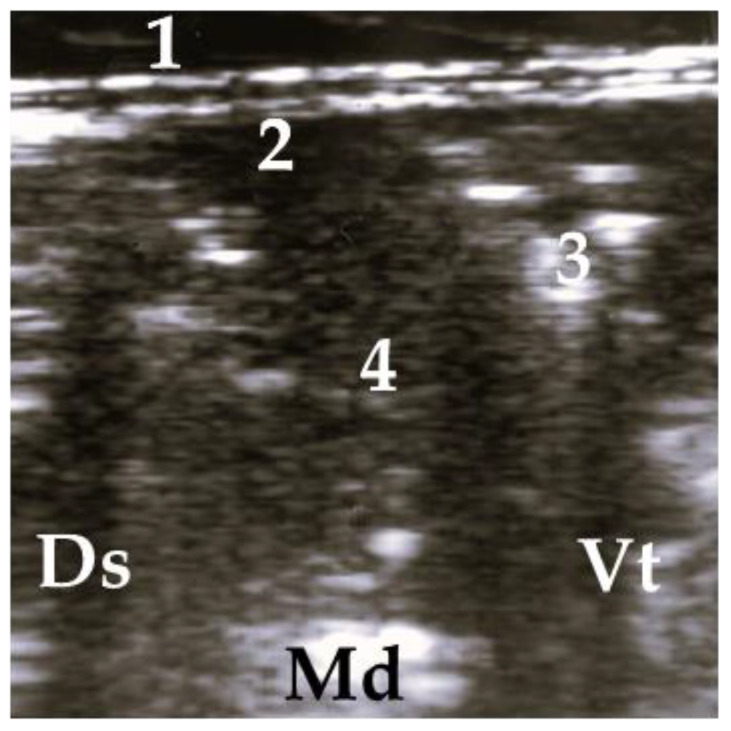
Lobar lesion of the whole lobe, 1 = thoracic wall, 2 = pleura, 3 = air bronchogram, 4 = consolidated lung; Ds = dorsal, Vt = ventral, Md = medial.

**Figure 6 animals-13-03464-f006:**
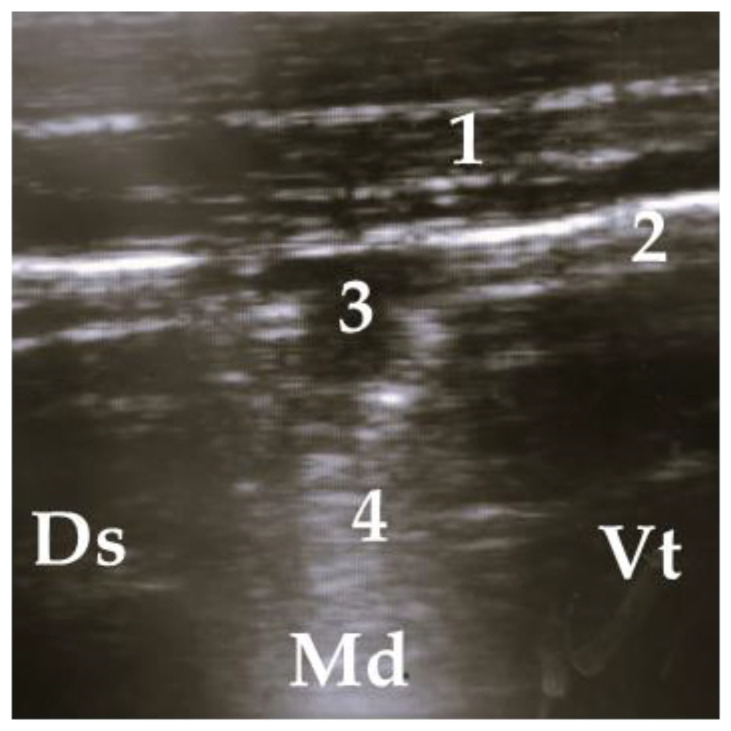
Superficial alveologram, 1 = thoracic wall, 2 = pleura, 3 = superficial alveologram, 4 = comet tail artifact; Ds = dorsal, Vt = ventral, Md = medial.

**Figure 7 animals-13-03464-f007:**
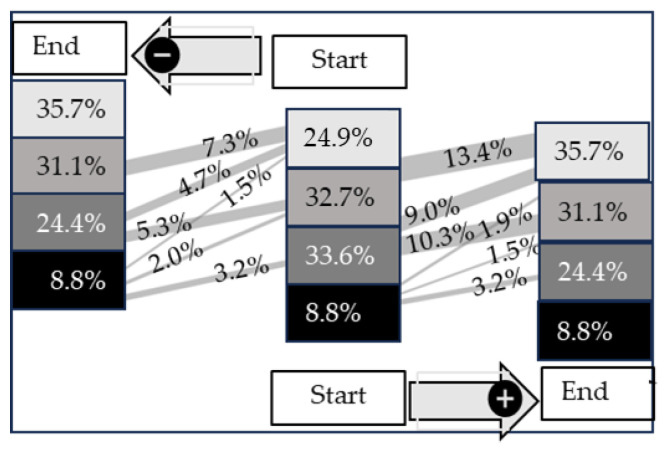
Sankey diagram of the development of COMT artifacts scores over time (start to end of fattening period) in 600 fattening calves. Lines from the start to the end of fattening indicate improvements (+) or deteriorations (−). The data are presented as %.

**Figure 8 animals-13-03464-f008:**
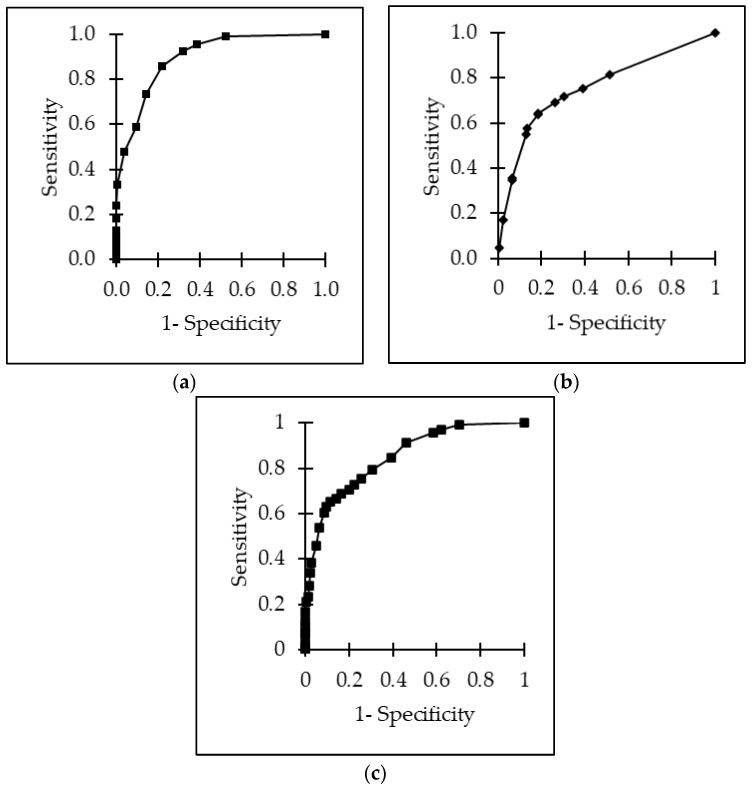
ROC curves for the detection of lung lesions using (**a**) the physical examination score, (**b**) the ultrasonographic examination score, and (**c**) a combined examination scoring of physical and ultrasonographic examination.

**Table 1 animals-13-03464-t001:** Bovine respiratory disease scoring system for pre-weaned dairy calves [9,23].

Diagnose ^1^	Description of the Clinical Sign	Score if Normal	Score if Abnormal (Any Severity)
ED	Serous, mucous, or purulent eye secretion; uni- or bilateral	0	2
ND	Serous, mucous, or purulent nose secretion; uni- or bilateral	0	4
HAT	slight uni- or bilateral ear droop or head tilt	0	5
CO	impulsive and audible exhalation sound	0	2
AB	Increased breathing frequency, signs of tachypnea	0	2
T		0 (<102.5 °F ≙ <39.2 °C)	2 (≥102.5 °F ≙ ≥39.2 °C)

^1^ Physical parameters: CO = cough, ND = nasal discharge, ED = eye discharge, AB = abnormal breathing, HAT = head tilt or ear droop, T = temperature.

**Table 2 animals-13-03464-t002:** Scoring system of the ultrasonographic examination [18,20].

Diagnose ^1^	Description	Score
COMTs and CON	No COMT, normal air content, no consolidation	0
	More than five COMT, diffuse allocated, no consolidation	1
	More than ten COMT, small lobular lesions	2
	Lobar lesion of the whole lobe	3
ALV	No ALV	0
	One or more ALV	1
ATA	No ATA	0
	One or more ATA	1
FLU	No FLU	0
	FLU	1

^1^ Ultrasonographic examination parameters: COMT = comet tail artifacts and CON = consolidation, ALV = alveologram, ATA = atelectasis, FLU = pleural effusion.

**Table 3 animals-13-03464-t003:** Scoring system of the postmortem examination modified by Leruste et al. (2012) [28].

Description	Score
Healthy lung with a normal pale orange color	0
One spot of grey-red discoloration	1
One larger or several small spots of grey-red discoloration	2
Grey-red discoloration area of a full lobe and/or presence of abscesses	3

**Table 4 animals-13-03464-t004:** Parameters of physical examination ^3^ in %. Data are provided for the seasons and the difference between the first examination (start) and the second examination (end). In total, 600 calves were examined (150 calves per season).

Season	Time of Examination	CO	ND	ED	AB	HAT	T	AU
Spring	Start	^3^ 3.1 ^b^	53.2 ^b,d^	30.0 ^b,c,d^	42.2	0.0	6.1 ^c,d^	55.5 ^d^
(*n* = 150)	Start − End ^2^	−17.11	−9.6	−15.9 ^1^	+11.6	±0.0	+29.2 ^1^	−11.9
Summer	Start	18.8 ^a,c,d^	43.6 ^d^	14.3 ^a,c^	43.6	0.0	9.0 ^c,d^	43.6 ^d^
(*n* = 150)	Start − End ^2^	−6.4	−8.9	−9.0 ^1^	−8.9	±0.0	−1.4	−14.21
Autumn	Start	21.1 ^d^	52.8 ^b,d^	38.2 ^d^	47.5	0.3	22.0 ^a,b,d^	68.6 ^a,b^
(*n* = 150)	Start − End ^2^	−2.0	−8.7	−19.1 ^1^	+3.8	−0.3	+5.6	−10.1
Winter	Start	32.3 ^b,c^	66.5 ^b^	13.0 ^a,c^	46.9	0.0	15.4 ^a^	69.7 ^a,b^
(*n* = 150)	Start − End ^2^	−13.21	−22.71	+1.1	+11.6	±0.0	+8.2	+12.9
Total	Start	26.0	53.8	24.7	45.2	0.1	13.6	41.2
(*n* = 600)	Start − End ^2^	−9.4	−12.4	−11.8 ^1^	+4.4	−0.1	+9.6	+12.8

^a–d^ Alphabetical indices for significant difference between seasons (*p* < 0.05): a = spring, b = summer, c = autumn, d = winter. ^1^ Numerous indices for significant difference between first and second examination (*p* < 0.05). ^2^ Difference between first and second examination in %. ^3^ Physical parameters: CO = cough, ND = nasal discharge, ED = eye discharge, AB = abnormal breathing, HAT = head tilt or ear droop, T = temperature, AU = auscultation.

**Table 5 animals-13-03464-t005:** Parameters of ultrasonographic ^2^ and postmortem examination in %. Data are provided for the seasons and the difference between the first examination (start) and the second examination (end). In total, 600 calves were examined (150 calves per season).

Season	Time of Examination	COMT/CON ^3^	ALV ^3^	ATA ^3^	FLU ^3^	1	2	3
Spring	Start	66.5 ^b^	4.9	3.0	0.8	14.0 ^a,d^	13.5 ^d^	8.3
(*n* = 150)	Start − End ^2^	+4.7	±0.0	+1.1	+0.7			
Summer	Start	80.5 ^a^	7.1	7.0	0.9	22.1 ^c,d^	6.9	4.7
(*n* = 150)	Start − End ^2^	−24.1 ^1^	−2.8	−1.7	−0.9			
Autumn	Start	76.3	2.5	1.4	0.8	6.6	8.1	4.2
(*n* = 150)	Start − End ^2^	+2.4	+1.0	+0.6	−0.1			
Winter	Start	70.1	3.3	3.2	0.6	2.6 ^a,b^	8.4 ^a^	8.1
(*n* = 150)	Start − End ^2^	−20.4	+0.6	−0.3	−0.6			
Total	Start	63.3	4.1	3.5	0.5	11.3	9.2	6.4
(*n* = 600)	Start − End ^2^	−10.5	−0.2	−0.4	−0.3			

^a–d^ Alphabetical indices for significant difference between seasons (*p* < 0.05): a = spring, b = summer, c = autumn, d = winter. ^1^ Numerous indices for significant difference between first and second examination (*p* < 0.05). ^2^ Difference between first and second examination in %. ^3^ Ultrasonographic examination parameters: COMT = comet tail artifacts and CON = consolidation, ALV = alveologram, ATA = atelectasis, FLU = pleural effusion.

**Table 6 animals-13-03464-t006:** Prevalence (%) of ultrasonographic examination parameters of the left lung-comet tail artifacts (COMT) and consolidations (CON), alveolograms (ALV), atelectases (ATA), pleural effusions (FLU) and findings of postmortem inspection. In total, 600 calves were examined (150 calves per season).

		Ultrasonography		Postmortem Inspection	
Lung Area (*n* = 600)	Diagnosis	Start	Start − End	Diagnosis	Prevalence
L1	COMT/CON	78.3	−13.01	0	80.5 ^b,c,d^
	ALV	4.0	−0.81	1	8.3 ^c,d^
	ATA	3.2	+0.2	2	6.2 ^c^
	FLU	0.7	+0.0	3	5.0 ^c^
	Total	21.5	−3.4	Total	19.5 ^b,c,d^
L2	COMT/CON	67.3 ^c^	−12.71	0	90.8 ^a,c^
	ALV	3.2	−0.8	1	3.0 ^c,d^
	ATA	2.8	−0.3	2	3.3 ^c,d^
	FLU	0.5	0.0	3	2.8 ^c^
	Total	18.5	−3.5	Total	9.2 ^a,c^

^a–d^ Alphabetical indices for significant difference between seasons (*p* < 0.05): a = spring, b = summer, c = autumn, d = winter.

**Table 7 animals-13-03464-t007:** Prevalence (%) of ultrasonographic examination parameters of the right lung—comet tail artifacts (COMT) and consolidations (CON), alveolograms (ALV), atelectases (ATA), pleural effusions (FLU), and findings of postmortem inspection. In total, 600 calves were examined (150 calves per season).

		Ultrasonography		Postmortem Inspection	
Lung Area (*n* = 600)	Diagnosis	Start	Start − End	Diagnosis	Prevalence
R1	COMT/CON	81.7 ^b^	−8.0	0	41.0 ^a,b,c,d^
	ALV	6.5	−1.0	1	20.5 ^a,b,d^
	ATA	4.5	+0.3	2	19.7 ^a,b,d^
	FLU	0.8	−0.3	3	15.8 ^a,b,d^
	Total	23.4	−2.3	Total	56.0 ^a,b,c,d^
R2				0	60.7 ^a,b,c,d^
				1	19.7 ^a,b,d^
				2	13.5 ^b,d^
				3	6.2 ^c^
				Total	39.3 ^a,b,c,d^
R3	COMT/CON	72.8	−9.3	0	89.2 ^c,d^
	ALV	5.5	−0.7	1	4.8 ^c,d^
	ATA	3.8	+0.3	2	3.2 ^c,d^
	FLU	0.8	−0.3	3	2.8 ^c^
	Total	20.8	−2.5	Total	10.8 ^c,d^

^a–d^ Alphabetical indices for significant difference between seasons (*p* < 0.05): a = spring, b = summer, c = autumn, d = winter.

## Data Availability

The data that support the findings of this study are available from the corresponding author upon reasonable request.

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
