# Peer review of "Diagnostic and Prognostic Value of Clinical Scoring and Lung Ultrasonography to Assess Pulmonary Lesions in Veal Calves"

_animals, 2023, doi:10.3390/ani13223464_

Round 1

Reviewer 1 Report

Comments and Suggestions for Authors

The work presented in this paper sounds interesting and deals with the subject that is one of important issues in the rearing period of veal calves. This period is critical not only for calves’ welfare reasons but also for the future productive value of these animals. This experiment is based on assessment of the association between an on-farm clinical scoring and lung ultrasonography with the post-mortem lungs inspection. The respiratory desease is considered one of the most important calves’ desease and its early detection and prevention are the key issues in the modern livestock production. The issues underlined in this manuscript provided a practical context for using tested examination methods for the detection of respiratory diseases in veal calves.

General comments:

The manuscript is clear, relevant for the field and presented in a well-structured manner. This manuscript has focused on two hypotheses: (i) clinical signs and ultrasonographic findings of lung diseases in veal calves at the beginning of the fattening period are associated with findings of thoracic ultrasonography at the end of the fattening period and (ii) the clinical scores and ultrasonographic findings at the end of fattening are well correlated to the lung pathology at slaughter. These two issues seem to be important to test effectiveness of diagnostic and prognostic value of used examination methods. The experimental design presented in the manuscript is appropriate to test both hypotheses.

The authors used a lot of abbreviations in their manuscript and it is a good idea to give the list of them at the end of the article. The cited references are relevant and mostly quite recent publications (within the last 10-12 years). The work is generally good, although some minor revisions could be done to improve the quality of the paper. Suggested changes are included in the specific comments.

Specific comments:

Line 147: AU is not mentioned in the six signs given in the Table 1, so it is not needed to have it in the legend to this table, however it is used in the line 150, and 'The AU'  can be explained as ‘The ausculation (AU)'.

 Conclusion section: the last sentence of the abstract:This study concluded that both physical and ultrasonographic examination scoring are reliable examination methods for the detection of lung diseases in veal calves.’ is worth to be mentioned (maybe in a bit modified version) also in conclusion section, especially that in the first conclusion sentence the authors underlined that ‘The limitations of ultrasonographic examination procedure must be considered when evaluating disease prevalence of BRD in veal calves.’

Comments on the Quality of English Language

Overall English language is appropriate and understandable, however there are just few words that need corrections (e.g. missed letter) to be done, for example: in the Table 1: ‘singn’ to ‘sign’, ‘doop’ to ‘droop’.

Author Response

Dear Reviewer 1!

Thank you for your respose! 

For research article

Response to Reviewer 1 Comments

1. Summary

2. Questions for General Evaluation

Reviewer’s Evaluation

Response and Revisions

Does the introduction provide sufficient background and include all relevant references?

Yes

Are all the cited references relevant to the research?

Yes

Is the research design appropriate?

Yes

Are the methods adequately described?

Can be improved

response in the point-by-point response letter.

Are the results clearly presented?

Can be improved

response in the point-by-point response letter.

Are the conclusions supported by the results?

Yes

3. Point-by-point response to Comments and Suggestions for Authors

Comment 1:

Line 147: AU is not mentioned in the six signs given in the Table 1, so it is not needed to have it in the legend to this table, however it is used in the line 150, and 'The AU can be explained as ‘The ausculation (AU)'.

Response 1:

Thank you, we agree with your comment. Therefore, we deleted “AU” from the legend (line 148). We explained the auscultation (AU) in line 150.

Comment 2:

Conclusion section: the last sentence of the abstract: This study concluded that both physical and ultrasonographic examination scoring are reliable examination methods for the detection of lung diseases in veal calves.’ is worth to be mentioned (maybe in a bit modified version) also in conclusion section, especially that in the first conclusion sentence the authors underlined that ‘The limitations of ultrasonographic examination procedure must be considered when evaluating disease prevalence of BRD in veal calves.

Response 2:

Conclusion: Both physical and ultrasonographic examination scoring are useful instruments to detect lung lesions in veal calves. Nevertheless, the limitations of ultrasonographic examination procedure must be considered when evaluating disease prevalence of BRD in veal calves.

Line 235- 238/ section 2

4. Response to Comments on the Quality of English Language

Point 1: Overall English language is appropriate and understandable, however there are just few words that need corrections (e.g. missed letter) to be done, for example: in the Table 1: ‘singn’ to ‘sign’, ‘doop’ to ‘droop’.

Response 1:

Thank you, we corrected the misspelled words in Table 1 and 4.

Page 4 and 10

Reviewer 2 Report

Comments and Suggestions for Authors

General comments:

The manuscript submitted by the authors Hoffelner Julia, Peinhopf-Petz Walter and Wittek Thomas describes a study on the diagnosis and prognosis of clinical scoring and lung ultrasonography to assess lung lesions in veal calves. The research is very interesting. However, some points should be taken into account to improve the quality of the paper.

Materials and Methods

Animals and Housing:

The authors point out that the animals studied came from different "markets in the Austrian federal states Styria and Salzburg, different breeds and ages". This suggests that groups of animals from different locations may have been treated differently (environmental conditions, housing, etc.). On the other hand, it is known that stress is an important factor in respiratory diseases. Therefore, I think it would be important (at least in the studies carried out at the beginning of the study) to differentiate or divide into groups of the same origin, breed, and age.

Results:

Tables 4 to 7 should indicate in number of alnimals (n) evaluated in each of the rows.

Comments on the Quality of English Language

Moderate editing of the English language is required

Author Response

Dear Reviewer!

Thank you for your response!

For research article

Response to Reviewer 2 Comments

1. Summary

2. Questions for General Evaluation

Reviewer’s Evaluation

Response and Revisions

Does the introduction provide sufficient background and include all relevant references?

Yes

Are all the cited references relevant to the research?

Yes

Is the research design appropriate?

Can be improved

response in the point-by-point response letter

Are the methods adequately described?

Can be improved

response in the point-by-point response letter

Are the results clearly presented?

Can be improved

response in the point-by-point response letter

Are the conclusions supported by the results?

Can be improved

response in the point-by-point response letter

3. Point-by-point response to Comments and Suggestions for Authors

Comment 1: Materials and Methods

Animals and Housing:

The authors point out that the animals studied came from different "markets in the Austrian federal states Styria and Salzburg, different breeds and ages". This suggests that groups of animals from different locations may have been treated differently (environmental conditions, housing, etc.). On the other hand, it is known that stress is an important factor in respiratory diseases. Therefore, I think it would be important (at least in the studies carried out at the beginning of the study) to differentiate or divide into groups of the same origin, breed, and age.

Response 1:

Thank you for making this point. We agree with this comment. Nevertheless, a preselection of the calves according to origin was not possible. In Austria the purchase of veal calves depends on cattle markets. A clear differentiation between calves from different houses of origin is not feasible, as most farms in Austria are small farms (average size 22 cows per farm) and do not follow standardized housing conditions. Nearly every calf arriving on the cattle marked originate from different farms. Dividing calves in groups of breeds was not possible due to high variables of pure and cross- breed calves. In both cases (origin and breed) the size of groups would be too small for a comprehensible statistical analysis. These are the typical limitation of a field study, we tried to discuss this and to reduce the impact of these factors by including a high number of calves in the study.

The influence factor of “age” at time of arrival (and body weight at arrival) on the daily weight gain at the fattening farm and treatment frequency will be discussed in a second publication.

The focus on this paper should be the ultrasonography and evaluation of clinical and ultrasonographic parameters during fattening.

Comments 2:

Tables 4 to 7 should indicate in number of animals (n) evaluated in each of the rows.

Response 2:

Table 4- 7 (page 10 and 11/section 1 and page 2/section 2): Agree. We revised the tables to emphasize this point. In total 600 calves were examined (150 calves per season).

4. Response to Comments on the Quality of English Language

Point 1: Moderate editing of the English language is required

Response 1:

Thank you. Correction of several misspelling words was done; We hope that we have been able to improve the English language.

Reviewer 3 Report

Comments and Suggestions for Authors

Good work and nicely presented. I have few comments and some questions:

- In line 128, please gloss BVD

- During the US examination, how do you determine the lobe you are scanning? Do you have any anatomical reference for that? Please clarify because your results are divided per lobe.

- I would include an example of ultrasound image for understanding the image description for the different findings.

- It would be nice if you remind to the reader that COMT/CON results could be considered as healthy animals. During the reading it can be unclear.

- In Figure 2, the central column sums 100.1. Please correct the values.

- In line 55, in results part, the US cut-off is 4 points, while physical exam is 6. But you don't say anything about the combination. Which number of points would be the threshold for considering the animal positive?

- In line 150, in discussion part, the prevalence of lung lesions is lower at slaughter than the exam parameters before. How do you explain it? Maybe overestimation of symptoms/images?

Author Response

Dear reviewer!

Thank you for your response!

Please see the attachment below!

Sincerely,

Julia Hoffelner
